# Atypical and Unique Transmission of Monkeypox Virus during the 2022 Outbreak: An Overview of the Current State of Knowledge

**DOI:** 10.3390/v14092012

**Published:** 2022-09-11

**Authors:** Jade C. Riopelle, Vincent J. Munster, Julia R. Port

**Affiliations:** Laboratory of Virology, Division of Intramural Research, National Institute of Allergy and Infectious Diseases, National Institutes of Health, Hamilton, MT 59840, USA

**Keywords:** monkeypox, transmission, outbreak, atypical presentation

## Abstract

An ongoing monkeypox outbreak in non-endemic countries has resulted in the declaration of a public health emergency of international concern by the World Health Organization (WHO). Though monkeypox has long been endemic in regions of sub-Saharan Africa, relatively little is known about its ecology, epidemiology, and transmission. Here, we consider the relevant research on both monkeypox and smallpox, a close relative, to make inferences about the current outbreak. Undetected circulation combined with atypical transmission and case presentation, including mild and asymptomatic disease, have facilitated the spread of monkeypox in non-endemic regions. A broader availability of diagnostics, enhanced surveillance, and targeted education, combined with a better understanding of the routes of transmission, are critical to identify at-risk populations and design science-based countermeasures to control the current outbreak.

## 1. Introduction

In May 2022, a case of monkeypox (MPX) was initially reported in the United Kingdom. As of 1 September 2022, this outbreak included over 52,090 confirmed MPX cases across 100 countries (Figure 1A) and was considered a public health emergency of international concern by the WHO. In contrast to historical MPX outbreaks in both Central and West Africa [1], the current outbreak has been able to sustain efficient human-to-human monkeypox virus (MPXV) transmission. All affected countries, as of 1 September 2022, were classified as non-endemic (Figure 1A), and many had never reported MPX cases before [2].

MPXV is a DNA virus in the *orthopoxvirus* (OPXV) genus; *orthopoxviruses* include the smallpox virus, which was eradicated through vaccination campaigns [3]. MPXV was first discovered in a 1958 outbreak amongst non-human primates (NHPs) in a Danish lab [4]. Human infection was documented for the first time in 1970 in the Democratic Republic of the Congo (DRC) [5]. MPX is endemic to several West and Central African countries [6] (Figure 1B).

Three phylogenetically distinct lineages comprise MPXV; historically, two have been recognized. These were referred to as the Central African (CA) and West African (WA) clades. These lineages are analogous to the two strains of smallpox, *Variola major* and *minor*, with regards to transmissibility, morbidity, and mortality. The updated nomenclature lists the former CA clade as clade 1, while clades 2A and 2B correspond to the WA clade [7]. Clade 1 MPXV has increased morbidity, mortality, viremia, and transmissibility [8] relative to clade 2. Case fatality rates (CFRs) across previously reported outbreaks have averaged 10.6% for clade 1 MPXV and 3.6% for clade 2 MPXV [5]. In humans, MPX presents with a 2–4-day prodrome followed by the appearance of a rash [9].

Historically, it has been thought that MPXV and smallpox are only transmissible after the onset of the rash and that subclinical infections with either virus are rare. However, the documentation of potentially asymptomatic MPXV infections, even before the current outbreak [10,11], challenges this. In the current outbreak, many patients are presenting without a prodromal phase and with mild or asymptomatic disease. The sustained human-to-human transmission seen in this outbreak has not been previously observed, highlighting the need for more information regarding the spread of MPXV. A comprehensive understanding of MPXV transmission requires an understanding of its ecology and spillover alongside the determinants of human-to-human transmission.

This review aims to compile and synthesize the current scientific understanding of MPXV’s ecology, with the goal of clarifying mechanisms of potential spillover, before focusing on various facets of MPXV’s transmission within the human population and finally contextualizing the current outbreak. The search criteria can be found in the Appendix A.

## 2. Ecology

The exact reservoir host complex of MPXV remains unknown, though two major candidates have been posited: giant pouched rats (*Cricetomys gambianus,* detection of OPXV antibodies) and rope squirrels (*Funisciurus* spp., detection of OPXV antibodies and virus isolation) [4,12,13,14,15]. Ecological niche models (ENMs) find the presence of rope squirrels, but not giant pouched rats, to be a significant predictor of MPXV’s geographical range [16,17]. Epidemiological studies have confirmed that human cases are elevated in areas predicted by ENMs to be ecologically suitable [18,19]. In the DRC, an increased prevalence of MPXV-specific antibodies was found in humans living in forested areas as opposed to the savannah [20].

Though its reservoir host is unknown, MPXV is thought to have a range of potentially suitable hosts, based on the presence of MPXV and OPXV antibodies, encompassing a wide variety of rodents and primates as dead-end hosts [13,14,21,22,23,24]. Though MPXV is the only known OPXV circulating in West and Central Africa, the possible presence of other OPXVs hampers definitive measurements of true MPXV prevalence. MPXV has only been successfully isolated from a dead sooty mangabey [25], a symptomatic rope squirrel [12], and the feces of a symptomatic chimpanzee [26]. Prior to the current outbreak, there had been no documentation of reverse spillover events [27] or outbreaks in domestic animals. However, the detection of OPXV antibodies in a domestic pig in the DRC [13] and up to 41% of peri-domestic rats in Uganda [28] underscores the zoonotic and cross-species potential of OPXVs.

## 3. Spillover

Outbreaks of MPX generally start with a spillover event preceding limited human-to-human transmission (Figure 2). Many potential routes of zoonotic transmission have been posited. In endemic areas, direct contact with animals, including dead or sick animals [9,29], and hunting, butchering, and eating bushmeat in particular [21,30,31,32,33,34], have been linked to infection. Bites and scratches have also been implicated [35], as has indirect transmission (i.e., via respiratory droplets). Highlighting the difficulties in elucidating potential routes of spillover are reports available from a 2003 outbreak in the U.S.A. An individual fell ill after a symptomatic prairie dog had been in their home, despite having no interaction with the animal and without the animal having purportedly touched any surface in the home [11], though a different study found no association between being in the vicinity of a sick prairie dog and MPXV infection [35]. Experimental studies demonstrating the successful aerosol inoculation of primates [36] as well as transmission studies in rope squirrels [22] and baboons [37] highlight the potential for aerosol, respiratory, and direct contact transmission from animals to humans.

MPX incidence in the DRC has increased 20-fold between 1981–1986 and 2006–2007 [38]. Simultaneously, countries considered by the WHO to be endemic for MPXV have recently expanded to include Nigeria and Cameroon, likely due to enhanced surveillance [39] and an increase in the susceptible population. Smallpox vaccination has been suggested to provide cross-protection against MPXV [40] but ended in the 1970s across much of sub-Saharan Africa, leaving up to 75% of the population unvaccinated [41]. Nigeria’s susceptible population is thought to have increased from 12.76 million (22.8%) in 1970 to 177.62 million (90.7%) in 2018 as a result of the cessation of smallpox vaccination and waning immunity in previously vaccinated populations [42]. Historically, most well-documented cases for which phylogenetic information is available have been infected with clade 1 MPXV, though there have been longitudinal increases in the documentation of infection with clade 2 MPXV. Most outbreaks have involved case counts in the single digits, but several recent outbreaks are thought to have been sustained by repeated spillover events [19,33,43] and nosocomial transmission [19,31,44] rather than ongoing human-to-human transmission (Figure 3).

Historically, primary cases resulting from spillover have disproportionately been young children, with boys comprising around 60% of primary cases [45,46]. This is likely the result of an inverse association observed between age and the likelihood of catching or eating rodents [47].

There have been instances of MPXV importation to non-endemic countries, with the most notable being a 2003 outbreak amongst prairie dogs and their human owners in the Midwestern United States [35,48] as well as several importations resulting from a 2017–2018 outbreak in Nigeria [49,50,51,52] (Figure 3).

## 4. Human-To-Human Transmission

### 4.1. Epidemiological Dynamics

Estimates of reproduction numbers vary widely for MPXV. An early estimate of 0.815 [40], based on MPX outbreak data in the DRC from 1980–1984, suggested that outbreaks are self-limiting in unvaccinated populations. However, recent mathematical modeling has revised this upward to as high as 2.13, indicating the potential for the ongoing circulation of MPXV in human populations [53].

In 2011–2012, the overall prevalence of anti-OPXV antibodies was 51% in Cote d’Ivoire and 60% in the DRC, likely driving the reproduction number of MPXV below one in these areas [54]. This is in accordance with the mathematical modeling of MPXV in populations with partial immunity: an estimate of the reproduction number in a population with 25% immunity was 1.10, indicating a reduced potential for prolonged outbreaks and circulation [53]. The seroprevalence of OPXV antibodies was up to 37% in those under age 23 (i.e., not vaccinated against smallpox) in Ghana, with children from rural forest communities significantly more likely to be seropositive [15]. Values of 19% in Cote d’Ivoire and 26% in the DRC [54] were observed in individuals born after 1985.

Based on the reproduction numbers, MPXV has historically been less transmissible than smallpox, for which convergent reproduction number estimates were around 4–6 [55]. The reproduction number for the prodromal period of smallpox was found to be 0.164 for a historical outbreak in Nigeria [56], corroborating the observation that smallpox patients were not particularly infectious before the onset of rash. MPXV has been thought to behave similarly.

Transmission heterogeneity has been documented for MPXV. It has been estimated that the top quintile of infectious patients, determined by transmission data from the DRC in 1980–1984, ultimately generate over 60% of subsequent cases [57], while the majority of primary cases fail to infect even one other person [10]: 67% of outbreaks in DRC involved only one case [40]. However, historical data may not accurately represent current trends. Despite the relative infrequency of transmission, there have been instances of superspreading events. In one outbreak, the likely index patient spread MPXV to eight family members [58], while in another incident in DRC, two children infected a total of eight people, none of whom transmitted MPXV onwards [44]. Historically, unrecognized or misdiagnosed illness has been the most important determinant of superspreading events [57]. Since the potential for unrecognized or misdiagnosed illness remains, MPXV transmission chain size may be an accurate predictor of potentially concerning outbreaks in the absence of detailed outbreak data [59].

### 4.2. Routes

In humans, MPXV shedding has been documented in urine, skin lesions [50], nasopharyngeal swabs, seminal fluid [60], blood [61], and feces [62]. Smallpox relied primarily on respiratory droplet transmission [63], with direct contact and fomite transmission playing less dominant roles [64]. For MPXV, it is thought that transmission via respiratory droplets, contact with bodily fluids or lesions, and contact with fomites are all possible [27] (Figure 2). Activities that specifically introduce MPXV to the oral mucosa (e.g., eating out of the same dish) are significantly associated with transmission, as opposed to events involving skin-to-skin contact (e.g., helping with bathing) [46]. The potential for vertical transmission may exist [65].

Given the observed airborne transmission of MPXV between animals in experimental settings, the detection of MPXV in upper respiratory samples [61], and the potential for the airborne transmission of smallpox [66,67], airborne human-to-human transmission of MPXV may be possible. However, epidemiological observations do not support airborne transmission as the primary route of transmission.

### 4.3. Determinants of Efficiency

Clade 2 MPXV has been thought to transmit less efficiently than clade 1 MPXV based on human data and animal models [68]. Existing rodent and NHP models demonstrated that an increased inoculation dose led to increased transmission, and more direct inoculation routes led to increased disease severity and transmissibility of MPXV [69,70]. In humans, complex exposures to MPXV-infected prairie dogs, defined as an invasive exposure (e.g., bite or scratch) combined with a non-invasive exposure (i.e., any exposure that did not break the skin), yielded a compressed disease progression, while non-invasive exposures only were associated with typical presentations of MPX [71]. These differences in disease progression and the potential for onward transmission likely influence epidemiological metrics such as generation time and reproduction number.

The proportion of those infected with MPXV that experience atypical or sub-clinical infection remains unclear, and the implications for the transmission of non-classically presenting infection are not understood. There has been infrequent or no documentation of the infection of contacts of smallpox patients during the prodrome [72] or in the absence of rash [73]. Though epidemiological observations [74] support the idea that MPXV shedding peaks with the appearance of the rash, the potential for transmission during the prodromal phase remains [74], with documented transmission to contacts of some patients in the pre-rash period [75].

Atypical MPX presentations resulting from non-traditional exposure routes may also make diagnosis difficult and increase the time from symptom onset (i.e., the putative start of infectiousness) to diagnosis. Even in settings with minimal documentation of unorthodox transmission routes, up to 13% of MPX cases might present atypically [9]. Based on historical experiences with the delayed diagnosis of smallpox and the resultant increase in transmission risk [57], MPXV transmitted via unusual routes, and thus presenting atypically, may prove more difficult to diagnose, resulting in larger outbreaks.

### 4.4. Risk Factors

Though specific risk factors vary between outbreaks, the importance of understanding the nuances of specific populations in predicting and anticipating outbreak dynamics cannot be overstated. Historically, MPX cases resulting from human-to-human transmission were more likely to be female, unvaccinated against smallpox, and living in the same residence and/or providing nursing care to a primary case [45]. Importantly, these data are based on clade 1 MPX cases in the DRC and may not reflect other endemic areas; documentation from outbreaks across endemic countries indicates that children bear much of the burden of MPX disease (Figure 3). In a recent outbreak of clade 2B MPX in Nigeria, 21–40-year-olds were primarily affected [27], though the index patient was an 11-year-old boy [50,76]. These risk factors indicate the role of behavioral and cultural determinants in facilitating the human-to-human transmission of MPXV.

Nosocomial MPXV transmission, to both patients and healthcare workers, remains a serious concern in outbreaks in both endemic and non-endemic regions (Figure 2). Smallpox was associated with nosocomial outbreaks [77], with the highest transmission rates occurring within hospitals [78]. Likewise, hospital-associated outbreaks of MPX are especially severe and long-lasting. This likely results from a combination of factors, including infections in vulnerable populations, hospital hygiene practice, and the use of aerosol-generating procedures [79]. Six generations of MPXV transmission were documented in a hospital in Impfondo, Republic of Congo, indicating MPXV’s potential to spread if not quickly addressed in healthcare settings [80]. In one incident in the United Kingdom, a healthcare worker who had handled the bedding and clothing of an MPX patient was infected with MPXV [75]. Precautions such as the use of appropriate personal protective equipment (PPE), proper waste management practices, and patient isolation should be implemented to minimize the hospital transmission of MPXV.

### 4.5. Countermeasures

There are two U.S. FDA-licensed vaccines against smallpox and monkeypox, ACAM2000 and JYNNEOS [81]. It has been posited that the cessation of regular smallpox vaccinations after its eradication has been a contributing factor to rising MPX cases [38]. Importantly, the JYNNEOS vaccine is a replication-deficient *vaccinia* virus vaccine, whereas ACAM2000 incorporates replication-competent live *vaccinia* virus; because of this, ACAM2000 is contraindicated for people living with HIV, regardless of immune status [82]. Replication-competent vaccines could cause clinical infection in humans as well as produce infectious virus that could be transmitted onwards, while replication-deficient vaccines do not produce infectious virus in humans and therefore pose a substantially lower risk of adverse events compared with replication-competent vaccines [83].

Populations at high risk for MPXV infection are often vaccinated prior to exposure. This has historically included lab workers and clinicians. Routine vaccination is not currently available in endemic countries. Post-exposure vaccination can reduce the risk of infection when given within 4 days of exposure and can reduce the severity of symptoms when given between 4 and 14 days after exposure [82]. However, the time between the onset of fever and the onset of rash has been shown to be longer, and the disease can present as mild or asymptomatic in vaccinated individuals, potentially altering transmission dynamics [84]. The extent of protection against MPXV breakthrough offered by these vaccines remains unclear.

There is currently no specific treatment approved for MPXV infection, though there are several antivirals developed to treat smallpox that are being tested, including tecovirimat, brincidofovir, and cidofovir. In a retrospective study of MPX cases in the United Kingdom from 2018 to 2021, one of seven patients was treated with tecovirimat and experienced a shorter duration of viral shedding [61], indicating that antivirals may help reduce the risk of MPXV transmission.

## 5. Current Outbreak

### 5.1. Epidemiology

The 2022 outbreak was first reported in the UK in May, and new cases were rapidly reported in other European countries. Though Europe and the U.S. remain the epicenters of the outbreak, cases have since been detected in Oceania, Asia, and elsewhere in the Americas [6,85] (Figure 1A).

The current outbreak has unfolded atypically in several capacities. CFRs have been low, continuous human-to-human transmission has been observed, and the outbreak remains in the exponential growth phase (Figure 4A). An estimate of the reproduction number in the early stages of this outbreak in Italy was 2.43 [86], with an observed decrease after 12 June. A later estimate based on all cases in this outbreak through 22 July 2022 was 1.29 [87]; both estimates highlight the ongoing exponential spread of MPXV. In contrast to historical importations of MPXV, the first detected cases in this outbreak have not been linked to endemic areas as of July 2022 [6]. Sequencing data indicate that this outbreak is caused by clade 2B MPXV and is subject to continuous microevolution (Figure 4B) [88]. Reverse spillover in the form of human-to-dog transmission has been documented for the first time in this outbreak [89].

Through the end of July 2022, 94% of U.S. cases occurred in men who reported recent male-to-male sexual or intimate contact (MSM) [94], with over 70% of cases in their 20s and 30s [95]. Given MPXV’s observed patterns of spread through sexual networks and documented instances of *vaccinia* virus transmission via sexual contact [96], it appears likely that sexual interactions at least partially contribute to the continued spread of MPXV in the current outbreak. MPXV DNA was found in seminal fluid at similar levels as shedding from nasopharyngeal swabs in some patients [60], and infectious MPXV has been isolated from seminal fluid [97]. However, the pattern of symptoms, especially lesion locations, indicates that sexual contact is the most likely route of transmission. MPXV DNA has persisted in inguinoscrotal lesions long after its clearance in other bodily fluids [61].

High-risk sexual behaviors, including unprotected sex and having sex with multiple anonymous or random sexual partners, are a risk factor in the current outbreak. In the early phase of this outbreak, men in whom MPX has been diagnosed had substantially higher rates of HIV infection than the general MSM population, with 14/27 confirmed cases in Portugal from 29 April to 23 May being HIV-positive [98] and an overall HIV positivity rate of 54·29% in a meta-analysis of 35 cases across five countries [95], signaling sexually promiscuous behavior. The dense sexual networks and high HIV prevalence rates of this MSM population are likely conducive to the continued spread of MPXV absent stronger public health measures.

Of U.S. cases through 22 July for which data were available, 41% were among non-Hispanic White, 28% were among Hispanic or Latino people, and 26% were among non-Hispanic Black people [94], indicating that racial and ethnic minorities were disproportionately affected. At the same time, non-Hispanic Black persons comprised only 15.6% of those receiving treatment, revealing barriers to accessing high-quality care for MPX [99]. Although the vast majority of cases are within the MSM population, there have also been reports of MPX cases in non-MSM populations [100].

Models predicting the course of the current outbreak are inconclusive. One model, assuming transmission within sexual networks and thus based on sexual partnership data in the United Kingdom, predicted a high likelihood of a major (>10,000 total cases) outbreak among the MSM population but low probability of sustained transmission in the non-MSM community in the absence of public health control measures [101]. However, a second SEIR (susceptible-exposed-infectious-recovered) model using the characteristics of a typical high-income European country but not assuming sexual transmission found that this outbreak should eventually subside, even in the absence of intervention. When accounting for public health measures, the second model found substantial reductions in outbreak size and duration [102]. Though the course of this outbreak remains uncertain, the second model, which does not heavily weigh the potential for sexual transmission in sustaining this outbreak, likely underestimates its expected size and duration. This discrepancy underscores the inability of mathematical models to adequately predict the progression of an outbreak in the early phase when real-life data on transmission routes, prevalence, and at-risk populations are missing.

### 5.2. Atypical Presentation

As stated, the misdiagnosis and underdiagnosis of MPX can lead to larger and longer outbreaks. In the current outbreak, there have been multiple reports of the initial misdiagnosis of patients who were later confirmed to have MPX [103,104] due to an atypical clinical manifestation that does not resemble the MPX observed in African outbreaks. There have been reports of patients presenting with no rash [105] and no prodrome [103]. In cases described in the United Kingdom, 20% of patients with a rash had no prodrome before rash onset, and only 11% of patients even presented with rash [105], the characteristic diagnostic marker for MPX. Estimates of the mean incubation period in the current outbreak have fallen in the lower range of the 7–14 day incubation period: 7.6 days based on patient data from the United States and the Netherlands [106] and 8.5 days based on cases only in the Netherlands [107].

Frequently in this outbreak, patients presenting atypically with genital or perianal ulcers have been initially misdiagnosed with common sexually transmitted infections (STIs) and sent home with antibiotics [60,103]. In the United States, the average time to diagnosis with an OPXV in the first 17 identified cases of this outbreak was 11 days after rash onset, with one patient seeking medical care four times over an eight-day period and another being diagnosed a full three weeks after the appearance of the rash [104]. Given the frequency of incorrect initial diagnoses, it is likely that the outbreak is larger than we currently believe and will continue to increase based on the historical patterns of MPXV and smallpox outbreaks. Furthermore, the observed trend of MPX patients repeatedly visiting healthcare facilities while actively contagious and undiagnosed increases the risk of healthcare-associated outbreaks.

Several factors likely contribute to the observed increase in atypically presenting MPX cases. Given that transmission routes might affect disease presentation, it is possible that this previously unrecognized mode of transmission results in clinical disease manifestations divergent from those previously observed. A recent retrospective study on a cohort with high rates of risky sexual behaviors from the 2017–2018 Nigeria outbreak indicated that 81.2% of them presented with genital ulcers [108]. The high rates of HIV coinfection may also contribute: one study found that HIV-positive MPX patients during the 2017–2018 outbreak in Nigeria had higher rates of genital ulcers and a higher likelihood of presenting with a genital rash as the first symptom than HIV-negative patients [109].

The less severe clinical disease caused by clade 2 MPXV may result in some of the less typically presenting cases observed in this outbreak. It is also possible that patients are less likely to seek medical care if they feel well. MPXV DNA has been found in semen samples of asymptomatic individuals, indicating that asymptomatic transmission might further facilitate the spread MPXV in this outbreak [110]. This is similar to the patterns noted in smallpox outbreaks: paradoxically, though *Variola minor* may be less inherently infectious than *Variola major*, *Variola minor* is actually associated with larger, longer, and less rapidly recognized outbreaks [72], likely as a result of its milder presentation leading to increased time to the diagnosis and isolation of cases. Likewise, infections with clade 2B MPXV in this outbreak have had long intervals from symptom onset to diagnosis, leading to increased transmission.

Especially in the early stages of this outbreak, the negative stigma associated with sexually promiscuous MSM communities or inexperience with MPX may have made providers more likely to immediately diagnose patients with routine STIs, prescribe antibiotics, and send them home rather than thoroughly considering the full range of potential differential diagnoses.

## 6. Conclusions

The burden of MPX has historically fallen primarily on a small number of sub-Saharan African countries, while the current outbreak has spread outside traditionally endemic areas and is disproportionately affecting MSM. We have a tenuous idea based on epidemiological observations, but a better understanding of fundamental questions in transmission will help determine evidence-based solutions to mitigate MPXV’s spread in both endemic and non-endemic areas. In both endemic and non-endemic areas, more in-depth surveillance and diagnostic methods will provide a richer understanding of the full extent of MPX cases, many of which are likely overlooked.

In general, high background levels of OPXV antibodies in MPXV-endemic areas suggest that many MPX cases go unrecognized. In Africa, MPX is a reportable disease through the integrated disease surveillance and response system [111], but reporting is likely uneven in the absence of readily available diagnostics. Likewise, the inability to pinpoint the source of the current outbreak or link cases to an endemic area indicates a similar pattern of undetected spread for a substantial period of time. Clearly, further efforts are needed to address the public health burden of MPXV. Differing strategies will need to be employed in endemic and non-endemic areas.

Active and syndromic surveillance methods would provide insight into fluctuations in epidemiological trends, especially in circumstances where confirmatory diagnostic methods are challenging. Regardless, it is recommended that the development of better diagnostics be prioritized, both point-of-care diagnostics that can be used outside of a healthcare facility as well as diagnostic approaches in healthcare settings, to reduce misdiagnosis. In particular, the success of rapid antigen tests during the SARS-CoV-2 pandemic provides a framework for scaling up the development and distribution of point-of-care diagnostics.

As children bear the burden of MPX cases in endemic areas, educating healthcare providers on the features that distinguish MPX from chickenpox will also help facilitate timely and accurate diagnoses, allowing for quicker isolation and behavior change. The targeted vaccination of healthcare workers or populations at risk for spillover events (e.g., children in forested areas) may be crucial to stemming outbreaks. Educating people on how to prevent spillover and protect children will be key to reducing both the incidence and size of outbreaks of both MPX and other zoonoses. Taking into consideration the current spread of MPXV in non-endemic countries and the social stigma and marginalization associated with being a member of the LGBTQ + community in endemic regions, it seems plausible that a similar spread in at-risk adult populations may occur undetected in these areas. Thus, safe access to inclusive healthcare could be crucial to address this gap.

We still lack clear information on specific routes of animal-to-human transmission as well as the range of potential reservoir hosts. The potential for a peri-domestic cycle of MPXV and the implications of such a cycle are also unknown. Furthermore, factors facilitating transmission from a primary case to the rest of the family are unclear. Finally, most historical research on MPXV has focused predominantly on data from clade 1 MPXV in the DRC, leaving the behavior of clade 2 MPXV less well understood. Further research on these topics will allow for more targeted education and prevention efforts that provide at-risk communities the opportunity to take action to lower the risk of spillover and ongoing transmission.

In the current outbreak, which as of now is primarily affecting sexually promiscuous MSM populations and may not be sustained in other communities, targeted approaches will be helpful. The vaccination of at-risk populations, including sex workers or those on PrEP, as a proxy for sexual promiscuity, might be crucial to preventing the further spread of this outbreak. Given that, in a recent study, only 44% of participants showed high intention to self-isolate after diagnosis with MPX [112], discussions about risky sexual behaviors and how to reduce risk could be helpful, especially in collaboration with community leaders. Given that many cases in this outbreak present atypically, expanding the case definition for MPX and educating healthcare providers on the broad range of clinical presentations of MPX as well as the features that distinguish MPX from common STIs will be crucial. Active surveillance methods will also capture a larger proportion of those with MPX, helping provide a more comprehensive understanding of the extent of this outbreak.

It remains unclear what proportion of cases in this outbreak present atypically and to what extent asymptomatic or atypical cases contribute to transmission. Furthermore, it is unknown whether MPXV’s spread in sexual networks results from contact with lesions during sex or true sexual fluid (i.e., seminal) transmission. Though MPXV is clearly spreading efficiently through networks of MSM, the transmission efficiency in heterosexual relationships or between women who have sex with women continues to be unclear. With the rapid increase in cases, the likelihood of transmission beyond networks of MSM increases dramatically. The observed microevolution of MPXV during the current outbreak is suggestive of a long unrecognized circulation of MPXV in the human population. Finally, the adaptation of MPXV to humans will likely result in more efficient replication and human-to-human transmission [88].

Addressing these questions will allow for more targeted public health interventions as well as a better understanding of whether this outbreak might be propagated in the non-MSM population. It is clear from the body of HIV/AIDS research that increased stigma towards those living with HIV is significantly associated with lower levels of medication adherence and usage of health services [113]. Working in conjunction with the media to send a message that though MSM have thus far been especially affected in the current outbreak, MPX is not a disease of MSM and being an MSM is not in itself a risk factor will be vital to reducing stigma. However, targeted interventions will be complicated in parts of the world where same-sex relationships are illegal [114]. At the same time, however, it is clear that targeted interventions stand to be the most beneficial. Working with established leaders in MSM communities and at-risk communities in endemic countries to establish mutually trusting relationships, with the goal of developing feasible and sensitive interventions, is key to addressing both the current outbreak of MPX and future MPX cases in endemic areas.

## Figures and Tables

**Figure 1 viruses-14-02012-f001:**
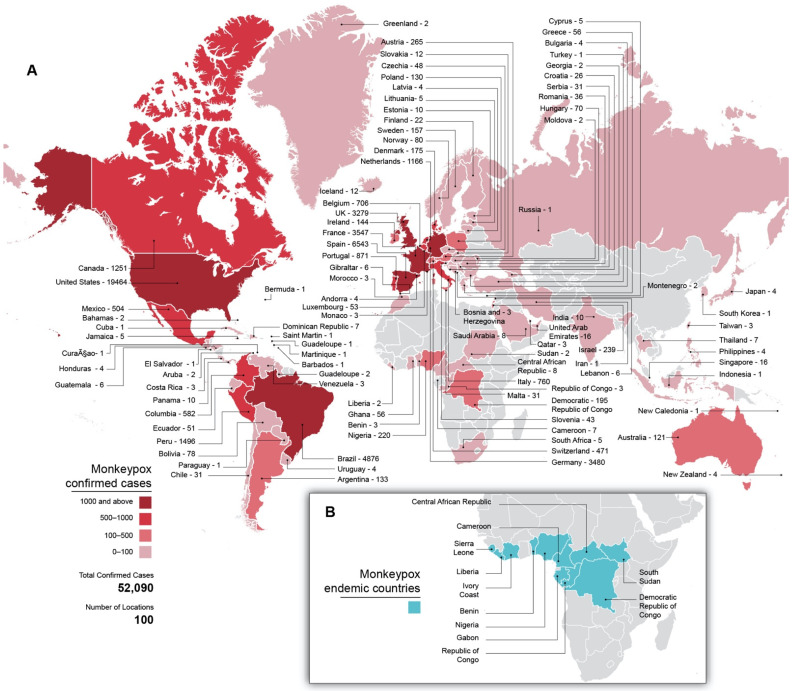
Global distribution of MPX cases. (**A**) MPX cases per country in the current outbreak through 1 September 2022 as well as total confirmed cases and affected countries. Countries are colored according to case counts. Gray shading indicates no known cases. (**B**) Countries historically endemic for MPXV. Gray shading indicates non-endemic countries.

**Figure 2 viruses-14-02012-f002:**
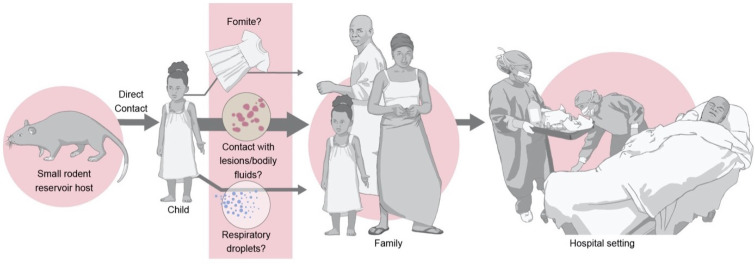
Posited routes of MPXV spillover and human-to-human transmission in endemic areas. The arrow thickness indicates the transmission likelihood.

**Figure 3 viruses-14-02012-f003:**
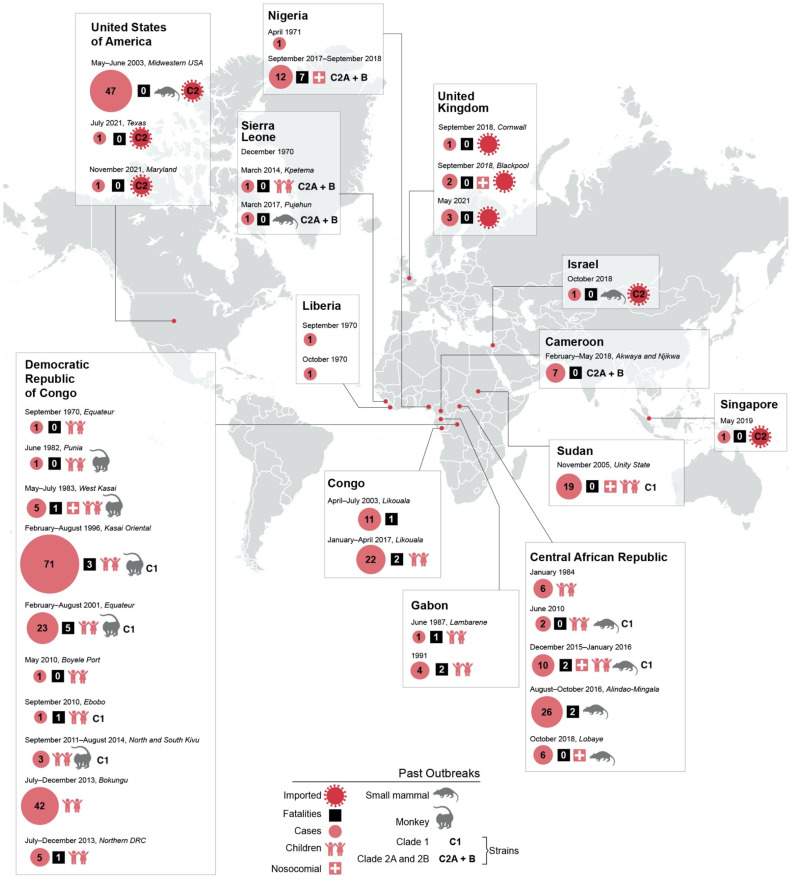
Historical outbreaks of MPX, including imported cases, organized by geographic location. Case numbers, fatalities, MPXV clade, and animal hosts are indicated where known. Outbreaks affecting primarily children and/or associated with nosocomial transmission are notated.

**Figure 4 viruses-14-02012-f004:**
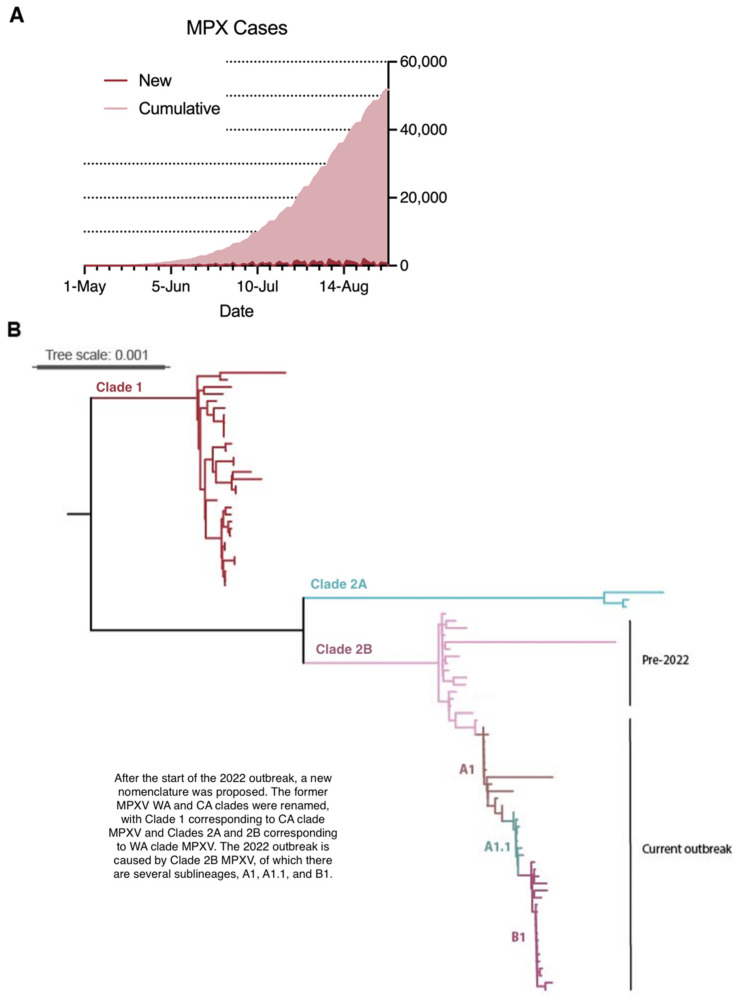
Epidemiology and phylogeny of the current outbreak. (**A**) Epidemic curve of daily new cases and cumulative total cases in non-endemic countries from 1 May 2022 to 1 September 2022. Data were retrieved from Global.health (https://github.com/globaldothealth/monkeypox) (accessed on 1 September 2022). (**B**) Phylogenetic trees showing sequences from the current outbreak alongside historical outbreaks. Full length genomes of all variants were obtained from GISAID database (https://www.gisaid.org/) (accessed on 18 July 2022) and GenBank (https://www.ncbi.nlm.nih.gov/genbank/) (accessed on 18 July 2022), and alignments were made with Muscle [90]. Substitution model was determined using ModelGenerator [91], and maximum-likelihood phylogenetic tree was constructed using IG-TREE2 [92] with substitution model F81 + F + I. Approximate likelihood ratio test (aLRT) was used to test branch supports (1000 replicates), and the tree was visualized in Itol [93] and midpoint-rooted for purposes of clarity. Only bootstrap values greater than 70% are shown. Bars indicate nucleotide substitutions per site. Clade and lineage are designated according to the nomenclature proposed by Happi et al. [7].

## Data Availability

Not applicable.

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
