# Peer review of "Atypical and Unique Transmission of Monkeypox Virus during the 2022 Outbreak: An Overview of the Current State of Knowledge"

_viruses, 2022, doi:10.3390/v14092012_

Round 1

Reviewer 1 Report

I read the paper titled “Atypical and unique transmission of monkeypox virus during the 2022 outbreak: an overview of the current state of knowledge,” by Jade C. Riopelle, Vincent J. Munster, and Julia R. Port, with great interest.

Overall, I found the manuscript poorly written. The following comments are in order:

(1) The writing style is too casual for a scientific publication. 

(2) Much of the relevant statistics is missing from the text, which simply narrates facts according to a strategy that remains obscure to the reader.

(3) Factual mistakes do appear in the text. For example, in the lines 48-50, the authors write: “Case fatality rates (CFRs) across previously reported outbreaks have averaged 3·6% for Clade 1 MPXV and 10·6% for Clades 2 and 3 MPXV (5).”, while, according to (5), the converse is true. 

(4) The figures look unprofessional. There is too much text in figure 1, that is hard to read, indicating that, in this case, a table would be more appropriate than a figure. Figure 2 contains more detail than required by the companion text.  In the legend of figure 3, “strains” should be “clades”. in figure 4A, the dynamic of the number of cases versus time is the most interesting data, yet hardly discernable.

(5) Section 4.1.: It is exceedingly clear that the authors missed many modeling papers in their search. They should in the very least discuss of the work by Blumberg and Lloyd-Smith, “Inference of R0 and Transmission Heterogeneity from the Size Distribution of Stuttering Chains”, PLoS Ccmput Biol 2013. Many other modeling papers would qualify for discussion within this section.

(6) Section 4.1.: The authors state without argument or citation: “MPXV remains less transmissible than smallpox”. If the authors cannot provide an argument, the statement should be deleted.

(7) I advise section 7 be deleted.

Author Response

Dear editor,

We appreciate the time and energy dedicated by the reviewers to providing suggestions for improving the manuscript. The incorporation of these insightful comments has improved the quality of this review. We would like to thank both the reviewers and the editor for their thorough consideration of our manuscript. We have also updated the review to the current state of knowledge for the current outbreak (Sep 2nd) and have added additional references as a result (indicated in red).

In-line answers to each point are found below:

Reviewer 1

Overall, I found the manuscript poorly written. The following comments are in order:

(1) The writing style is too casual for a scientific publication.

            We thank the reviewer for their thoughtful feedback. Though this review is primarily intended to inform non-subject matter experts (e.g., other scientists, clinicians, and even the general public) on the pressing topic of MPXV transmission, we appreciate that the verbiage may occasionally tend towards the colloquial as a result. With this feedback in mind, we have revised passages that are especially conversational. These include lines 145-147, 156-157, and 495, among others. Changes of verbiage are indicated throughout the text.

(2) Much of the relevant statistics is missing from the text, which simply narrates facts according to a strategy that remains obscure to the reader.

            We value the insight afforded to us by this comment. In an attempt to clarify the goals and organizational structure of this manuscript to future readers, we have added further information at the end of the introduction, as follows:

“This review aims to compile and synthesize the current scientific understanding of MPXV’s ecology with the goal of clarifying mechanisms of potential spillover before focusing on various facets of MPXV’s transmission within the human population and finally using our findings to contextualize the current outbreak.” (Lines 61-64).

Furthermore, we have included a section on our search strategy and selection criteria of manuscripts reviewed and compiled (previously section 7), which provides further information on the methods we used in developing and organizing this manuscript. At the reviewer’s suggestion, we have removed this as a separate section and have moved the relevant information to the supplement to clearly state our methodology.

            Though we are uncertain as to what is meant by “relevant statistics”, we have interpreted it to indicate that we should provide further support with numerical data where applicable. With this in mind, we have added quantitative data to the manuscript when possible to further support our claims. Specific examples include:

  • “Smallpox vaccination has been suggested to provide cross-protection against MPXV but ended in the 1970s across much of Sub-Saharan Africa, leaving up to 75% of the population unvaccinated.” (Lines 110-112)
  • “Nigeria’s susceptible population is thought to have increased from 12.76 million in 1970 to 177.62 million in 2018…” (Lines 112-114)
  • “…with boys comprising around 60% of primary cases.” (Line 123)
  • “Through the end of July 2022, 94% of U.S. cases occurred in men who reported recent male-to-male sexual or intimate contact, with over 70% of cases in their 20s and 30s.” (Lines 300-302)

(3) Factual mistakes do appear in the text. For example, in the lines 48-50, the authors write: “Case fatality rates (CFRs) across previously reported outbreaks have averaged 3·6% for Clade 1 MPXV and 10·6% for Clades 2 and 3 MPXV (5).”, while, according to (5), the converse is true.

We have since corrected this error and have found no further factual errors after additional scrutiny of the manuscript.

(4) The figures look unprofessional. There is too much text in figure 1, that is hard to read, indicating that, in this case, a table would be more appropriate than a figure. Figure 2 contains more detail than required by the companion text.  In the legend of figure 3, “strains” should be “clades”. in figure 4A, the dynamic of the number of cases versus time is the most interesting data, yet hardly discernible.

            The figures were designed by a professional medical illustrator, and we believe them informative to the reader, readily interpretable, and suitable for publication in their current format. Though we sincerely value the insight and respect the opinion of the reviewer, we prefer to maintain the current format of Figure 1. We believe that displaying case counts from the current outbreak as a color shaded map provides a visually appealing and easy way to allow readers to process and understand the large volume of information. Recognizing that some readers may prefer a table, we have also created a table displaying the same information, which is included as supplemental information.

            All the information displayed in Figure 2 is also mentioned in the manuscript text, and we believe that highlighting potential transmission routes in endemic areas provides important context for the current outbreak. To make this information easier for the reader to find, we have explicitly referenced Figure 2 in appropriate locations in the text that discuss information shown in the figure. Relevant references occur in lines 139, 278, and 344. We agree with the reviewer that Figure 2B does not provide additional clarity and have removed this figure.

We appreciate the attention to detail that the reviewer paid to Figure 3. Accordingly, we have updated the labels in Figure 3 to reference “Clade 1” and “Clade 2”.

            Figure 4A has been updated for ease of understanding and interpreting the data. Relevant changes include changing the position of the y axis for clearer interpretation of recent case counts, the addition of horizontal lines providing visual benchmarks at certain numbers, and changing the x axis labels to clearly show the earliest and latest dates included in these data. Figure 4A has also been updated to include the most recent case counts.

(5) Section 4.1.: It is exceedingly clear that the authors missed many modeling papers in their search. They should in the very least discuss of the work by Blumberg and Lloyd-Smith, “Inference of R0 and Transmission Heterogeneity from the Size Distribution of Stuttering Chains”, PLoS Ccmput Biol 2013. Many other modeling papers would qualify for discussion within this section.

            We thank the reviewer for their efforts to help us incorporate more relevant modelling literature into this manuscript. Though we recognize the crucial role of mathematical models in helping medical professionals, public health officials, and policymakers understand and control the spread of infectious diseases, the aim of this review was to provide a broad overview of many different aspects of transmission, not a specific focus on models proposed for the reproductive number of MPXV. We have added additional information clarifying our goals in this manuscript (see point 2). At the same time, appreciating the value of including all perspectives in our work, we have conducted an additional thorough search for further mathematical modelling work, which yielded one additional paper (Grant et al. 2020) that has since been added to Section 4.1.

(6) Section 4.1.: The authors state without argument or citation: “MPXV remains less transmissible than smallpox”. If the authors cannot provide an argument, the statement should be deleted.

            We appreciate the suggestion and have rephrased this claim for clarity. It now reads: “Based on reproduction numbers, MPXV has historically been less transmissible than smallpox, for which convergent reproduction number estimates were around 4-6” (Lines 153-154), referencing both the reproduction numbers for MPXV in the preceding paragraph and reproduction number estimates for smallpox.

(7) I advise section 7 be deleted.

We value the suggestion to delete the section on our search strategy and selection criteria. Believing that it provides important context on our work and the resultant conclusions, we have instead chosen to include it as supplemental information.

Reviewer 2 Report

A fine and timely review. Nice figures too.

Just a couple of points where I think you could increase clarity for a reader:

L44 You say "These lineages are analogous to the two strains of smallpox, Variola major and minor". It's not clear in what sense you mean this; genomes? symptoms? transmissibility? I don't think this statement is needed, but if kept I would advise rewording. I know you return to this point at the end, but this statement should make sense in this context.

L49 Am I misreading something here? You say Clade 1 has increased mortality, and then describe the CFR as 3.6 for Clade 1 compared to 10.6 for Clade 2 and 3.

L63 I think it would be easier to read by swapping the two paragraphs in '2. Ecology'. The reason is that you firstly describe/imply rodents and primates as dead-end hosts, when the likely host is actually a rodent (as you spell out in the second paragraph).

L84 The way you've worded it here, you seem to be saying that hunting and butchering is not direct contact.

L94 Not sure what you mean by "highlight different potential routes of spillover". I think you need to say exactly what you mean here; i.e. fomite, direct contact, aerosol.

L97 Panel B kind of implies heterosexual sex.

L103 and L114 There's nice data on this and the change in vaccinated population as the population ages. A couple of numbers would be nice here, rather than just 'increase' and 'large'.

L127 It's a little hard to follow what you're saying in the first paragraph of '4.1 Epi Dynamics'. You go from self-limiting un unvaxxed, to low R due to OPXV Ab (maybe you can explicitly say this is vaccination for VARV?), and back to non-vaxxed (again, maybe be explicit that here OPXV Ab are from endogenous MPXV infection, is that the inference?).

L165 Important to say this is experimental animals. Or was there evidence for airborne in the transported prairie dogs in the 2003 US outbreak? Maybe clarify what you mean here.

L177 What is non-invasive exposure?

L280 There are estimates on what proportion of cases in the current outbreak are in the MSM community, most of them are thought to be underestimates as well. Something like 90-95% Putting some numbers here is important.

L305 I'm not clear on your point here. Both models are in the absence of intervention, but you make it sound like this is the key aspect of the second model?

Author Response

Dear editor,

We appreciate the time and energy dedicated by the reviewers to providing suggestions for improving the manuscript. The incorporation of these insightful comments has improved the quality of this review. We would like to thank both the reviewers and the editor for their thorough consideration of our manuscript. We have also updated the review to the current state of knowledge for the current outbreak (Sep 2nd) and have added additional references as a result (indicated in red).

In-line answers to each point are found below:

Reviewer 2

L44 You say "These lineages are analogous to the two strains of smallpox, Variola major and minor". It's not clear in what sense you mean this; genomes? symptoms? transmissibility? I don't think this statement is needed, but if kept I would advise rewording. I know you return to this point at the end, but this statement should make sense in this context.

            We appreciate the reviewer’s feedback on clarifying our language. We have rephrased this sentence such that it now reads: “These lineages are analogous to the two strains of smallpox, Variola major and minor, with regards to transmissibility, morbidity, and mortality,” (Lines 45-46).

L49 Am I misreading something here? You say Clade 1 has increased mortality, and then describe the CFR as 3.6 for Clade 1 compared to 10.6 for Clade 2 and 3.

We thank the reviewer for their close reading of our manuscript. We have since corrected this error and have found no further factual errors after additional scrutiny of the manuscript.

L63 I think it would be easier to read by swapping the two paragraphs in '2. Ecology'. The reason is that you firstly describe/imply rodents and primates as dead-end hosts, when the likely host is actually a rodent (as you spell out in the second paragraph).

            We have reversed the order of the paragraphs in this section and value the additional clarity this change provides.

L84 The way you've worded it here, you seem to be saying that hunting and butchering is not direct contact.

            We value the reviewer’s insightful feedback. We have rephrased this section for clarity – this sentence now says: “In endemic areas, direct contact with animals, including dead or sick animals, and hunting, butchering, and eating bushmeat in particular, have been linked to infection,” (Lines 91-93).

L94 Not sure what you mean by "highlight different potential routes of spillover". I think you need to say exactly what you mean here; i.e. fomite, direct contact, aerosol.

            We have clarified this clause so that it now reads “highlight the potential for aerosol, respiratory, and direct contact transmission from animals to humans,” (Lines 101-102).

L97 Panel B kind of implies heterosexual sex.

            After careful consideration of both reviewers’ feedback and the information included in the body text of this manuscript, we have decided to remove Figure 2B entirely to avoid confusion and misinterpretation.

L103 and L114 There's nice data on this and the change in vaccinated population as the population ages. A couple of numbers would be nice here, rather than just 'increase' and 'large'.

            We thank the reviewer for their thoughtful comments. In response, we have added appropriate numerical values to the manuscript. In particular, we have incorporated data on Nigeria’s susceptible population (“Nigeria’s susceptible population is thought to have increased from 12.76 million in 1970 to 177.62 million in 2018”, Lines 112-114) and the proportion of the population that is unvaccinated (“leaving up to 75% of the population unvaccinated”, Line 112).

L127 It's a little hard to follow what you're saying in the first paragraph of '4.1 Epi Dynamics'. You go from self-limiting un unvaxxed, to low R due to OPXV Ab (maybe you can explicitly say this is vaccination for VARV?), and back to non-vaxxed (again, maybe be explicit that here OPXV Ab are from endogenous MPXV infection, is that the inference?).

            We value the reviewer’s perspective and have added a sentence that we hope allays the confusion regarding this paragraph. We find that “recent mathematical modelling has revised [the reproduction number] upward to as high as 2.13” (Lines 141-142) before discussing factors that may drive the reproduction number below one in certain areas.

L165 Important to say this is experimental animals. Or was there evidence for airborne in the transported prairie dogs in the 2003 US outbreak? Maybe clarify what you mean here.

            We appreciate the insight afforded to us by this comment. We have rephrased to clarify that airborne transmission has only been observed in an experimental setting (Lines 183-184).

L177 What is non-invasive exposure?

            This request for clarity has been noted and accounted for. The manuscript now states that non-invasive exposures include “any exposure that did not break the skin” (Lines 195-196).

L280 There are estimates on what proportion of cases in the current outbreak are in the MSM community, most of them are thought to be underestimates as well. Something like 90-95% Putting some numbers here is important.

            We recognize the value of attaching numerical estimates to our claims. In response, we have added that “94% of U.S. cases occurred in men who reported recent male-to-male sexual or intimate contact” (Lines 300-301) in an attempt to provide more concrete details.

L305 I'm not clear on your point here. Both models are in the absence of intervention, but you make it sound like this is the key aspect of the second model?

            We have rephrased our paragraph on models to hopefully provide more structure and clarity (Lines 331-332, 335, 337) – we aimed to emphasize that the difference between the two models is that one accounts for sexual transmission and the other does not.

Round 2

Reviewer 1 Report

-